# UAV-DETR: An Enhanced RT-DETR Architecture for Efficient Small Object Detection in UAV Imagery

**DOI:** 10.3390/s25154582

**Published:** 2025-07-24

**Authors:** Yu Zhou, Yan Wei

**Affiliations:** College of Computer and Information Science, Chongqing Normal University, Chongqing 401331, China; 2023110516055@stu.cqnu.edu.cn

**Keywords:** small object detection, UAV imagery, RT-DETR, feature fusion

## Abstract

To mitigate the technical challenges associated with small-object detection, feature degradation, and spatial-contextual misalignment in UAV-acquired imagery, this paper proposes UAV-DETR, an enhanced Transformer-based object detection model designed for aerial scenarios. Specifically, UAV imagery often suffers from feature degradation due to low resolution and complex backgrounds and from semantic-spatial misalignment caused by dynamic shooting conditions. This work addresses these challenges by enhancing feature perception, semantic representation, and spatial alignment. Architecturally extending the RT-DETR framework, UAV-DETR incorporates three novel modules: the Channel-Aware Sensing Module (CAS), the Scale-Optimized Enhancement Pyramid Module (SOEP), and the newly designed Context-Spatial Alignment Module (CSAM), which integrates the functionalities of contextual and spatial calibration. These components collaboratively strengthen multi-scale feature extraction, semantic representation, and spatial-contextual alignment. The CAS module refines the backbone to improve multi-scale feature perception, while SOEP enhances semantic richness in shallow layers through lightweight channel-weighted fusion. CSAM further optimizes the hybrid encoder by simultaneously correcting contextual inconsistencies and spatial misalignments during feature fusion, enabling more precise cross-scale integration. Comprehensive comparisons with mainstream detectors, including Faster R-CNN and YOLOv5, demonstrate that UAV-DETR achieves superior small-object detection performance in complex aerial scenarios. The performance is thoroughly evaluated in terms of mAP@0.5, parameter count, and computational complexity (GFLOPs). Experiments on the VisDrone2019 dataset benchmark demonstrate that UAV-DETR achieves an mAP@0.5 of 51.6%, surpassing RT-DETR by 3.5% while reducing the number of model parameters from 19.8 million to 16.8 million.

## 1. Introduction

The accelerated evolution of drone technologies and sustained improvements in data transmission capabilities have revealed significant utility for UAVs in multiple domains, including surveillance, emergency rescue, and supply chain management. The capacity for drones to acquire high-resolution remote sensing images with ease is facilitated by their advanced data acquisition capabilities, providing significant support for target detection tasks. Consequently, the precise identification of targets utilizing UAV-captured imagery has garnered substantial scholarly attention.

Since AlexNet [1] introduced Convolutional Neural Networks (CNNs) and achieved remarkable success in the ImageNet competition in 2012, CNN-based approaches have rapidly become the mainstream in the field of object detection. Subsequently, two-stage detectors, such as the R-CNN series [2,3], were proposed, which significantly improved both detection accuracy and speed. However, their high computational cost makes them unsuitable for deployment on UAV platforms. To address the limitations of two-stage detectors in terms of real-time performance, one-stage detectors such as YOLO [4] and SSD [5] were developed. Later studies focused on improving detection accuracy by addressing the specific challenges inherent in aerial imagery. For example, Sahin et al. [6] enhanced the detection of small objects in aerial images by expanding YOLOv3’s output layers to five, which improved accuracy but at the cost of significantly increased parameters. Similarly, Tan et al. [7] introduced the Receptive Field Block (RFB) module into YOLOv4 to address issues such as severe occlusion and uneven sample distribution in aerial scenarios. However, the use of dilated convolutions within the RFB module may introduce noise information. Although these modifications have effectively mitigated some challenges associated with aerial images, the intrinsic limitations of CNNs in capturing long-range dependencies render them suboptimal for aerial image-based tasks.

With the remarkable success of the Transformer architecture [8] in the field of natural language processing, researchers have begun exploring its innovative concepts for object detection tasks. In 2020, the Facebook team proposed DETR [9], an end-to-end object detection framework based on Transformers, which elegantly eliminates the need for the aforementioned hand-crafted processing steps, thereby streamlining the detection pipeline. Although DETR offers a more efficient and direct detection paradigm, its substantial parameter count poses a significant challenge, necessitating strategies for parameter optimization while maintaining model performance. As a result, numerous DETR variants have been proposed to address the issues of large parameter size and high computational cost. In 2021, Zhu et al. introduced Deformable DETR [10], which enhances the attention mechanism by focusing solely on key sampling points around reference locations to improve performance. In 2022, Roh et al. proposed Sparse DETR [11], which reduces the attention complexity within the encoder through a token sparsification strategy, thereby improving detection performance under the same computational budget. Also in 2022, Li et al. proposed DN-DETR [12], which leverages denoising training to address the instability of bipartite matching in DETR, significantly accelerating model convergence. Furthermore, Li et al. introduced DINO [13] in 2022, which achieves substantial model size reduction by improving both the encoder and decoder structures. To address the high computational overhead of DETR, Baidu proposed RT-DETR [14], a real-time detection framework based on DETR that removes threshold filtering and non-maximum suppression (NMS), enabling RT-DETR to achieve higher training accuracy with fewer iterations.

Although RT-DETR has achieved promising results, its neck network still relies on the traditional pyramid structure, which fails to fully exploit the correlations between feature maps at different scales, leading to suboptimal feature fusion. Moreover, UAV aerial images often suffer from uneven sample distribution and severe occlusions. The fixed receptive field of convolutional kernels in the neck network is insufficient to capture both global context and fine-grained details simultaneously. Therefore, based on RT-DETR, an improved UAV-DETR detector is proposed to achieve a balance between detection accuracy and computational efficiency. The main contributions of this work are as follows:A lightweight and efficient backbone is developed by introducing the Channel-Aware Sensing (CAS) module, which replaces conventional convolutional blocks with a spatial-channel attention mechanism. This architecture not only enhances small-target detection capability but also substantially decreases parameter volume and computational demands. Experiments show that, compared with the original ResNet-18 backbone, CAS achieves better detection accuracy with lower GFLOPs and fewer parameters.The Scale-Optimized Enhancement Pyramid (SOEP) is a proposed model that incorporates a small-object detection layer into the original multi-scale feature fusion network. It introduces the concept of Cross Stage Partial (CSP) and OmniKernel network and designs the CSP-OmniKernel module for feature fusion to enhance efficiency in feature representation learning. The CSP-OmniKernel module is engineered to facilitate feature fusion for enhanced detection performance of small targets, ranging from global to local perspectives.A unified Context-Spatial Alignment Module (CSAM) is introduced, integrating Contextual Fusion and Calibration (CFC) and Spatial Feature Calibration (SFC) to jointly address semantic inconsistency and spatial misalignment in cross-scale feature fusion.Extensive experiments on the VisDrone2019 dataset demonstrate that the proposed UAV-DETR achieves an mAP@0.5 of 51.6%, outperforming the baseline RT-DETR by 3.5%, while maintaining low computational complexity suitable for real-time UAV-based applications.

## 2. Related Work

### 2.1. CNN-Based Object Detection

Convolutional Neural Networks (CNNs) have laid the foundation for modern object detection. Girshick et al. [2] proposed R-CNN, which introduced region proposals and convolutional feature extraction, later improved by Fast R-CNN and Faster R-CNN. Redmon et al. [4] pioneered the YOLO (You Only Look Once), achieving real-time inference by integrating object localization and classification within an end-to-end unified network architecture. Liu et al. proposed SSD [5], which leveraged multi-scale features for efficient detection. These CNN-based models have been widely applied in drone detection tasks due to their balance between speed and accuracy.

However, UAV imagery presents unique challenges such as small, densely packed targets, large-scale variations, and complex backgrounds. To mitigate these, Lin et al. proposed Feature Pyramid Networks (FPN) [15] to enhance multi-scale feature extraction. Subsequent improvements like PANet [16] and BiFPN [17] further strengthened feature aggregation and semantic consistency across levels. Nonetheless, CNNs still struggle to capture global context due to their limited receptive fields.

### 2.2. Transformer-Based Object Detection

Transformer-based detectors offer significant potential for aerial analysis, where their global attention mechanisms effectively correlate spatially dispersed objects across wide-area UAV views, overcoming the limited receptive fields of CNNs. Carion et al. [9] introduced Detection Transformer (DETR). This approach reformulates object detection as a set prediction task, effectively eliminating the reliance on hand-crafted components such as anchor boxes and non-maximum suppression (NMS). Despite its architectural simplicity and end-to-end paradigm, DETR exhibits significant limitations: (1) slow convergence requiring 500 training epochs to achieve competitive accuracy; (2) suboptimal performance on small objects due to an information bottleneck in global attention mechanisms.

To address these limitations, Deformable DETR by Zhu et al. [10] employs sparse attention across a few key sampling points to speed up training and improve accuracy. Conditional DETR [18] and DAB-DETR [19] further refined the query representation and attention mechanisms. Li et al. proposed DN-DETR [12], incorporating denoising techniques to improve convergence stability.

In the real-time detection domain, RT-DETR was proposed by Baidu’s AI team, offering a hybrid encoder with intra-scale and cross-scale interaction modules, along with an IoU-aware query initialization mechanism. An effective trade-off between detection speed and accuracy is obtained, making it well-suited for edge deployment. However, when directly applied to UAV scenarios, RT-DETR still struggles with issues such as feature degradation from repeated downsampling, small object miss detection, and spatial-context misalignment.

### 2.3. UAV-Specific Detection Methods

Several works have specifically optimized detection for UAV scenarios. Yang et al. proposed UAV-YOLO [20], which improves small object sensitivity through multi-resolution feature fusion. Guo et al. introduced TinyDet [21], targeting low-computation platforms. Datasets such as VisDrone [22], UAVDT [23], and DOTA [24] have become benchmarks for evaluating UAV detection models. Nevertheless, many existing models either sacrifice accuracy for speed or fail to generalize across complex UAV scenes with scale, motion, and background variability.

While significant progress has been made from CNN-based to Transformer-based detection frameworks, challenges such as small object detection, feature misalignment, and efficient multi-scale fusion persist in UAV applications. Several recent DETR variants, such as Efficient-DETR and SMCA-DETR, have attempted to improve training convergence and localization accuracy. Moreover, recent approaches such as CFPT (Cross-Layer Feature Pyramid Transformer) [25] and LGI-DETR (Local–Global Interaction DETR) [26] have further explored cross-layer attention mechanisms and local-global feature fusion to mitigate spatial-contextual misalignment in UAV imagery. However, these methods often incur substantial architectural complexity, with parameter counts reaching 51.3 million and 21.1 million, respectively, and rely heavily on decoder-side enhancement modules. This results in significant computational overhead, thereby limiting their real-time performance and applicability in resource-constrained environments. Recent studies have also made notable progress in small object detection on the VisDrone2019-DET dataset. Zhao et al. [27] proposed an improved YOLOv7 model that achieves a high mAP@0.5 of 52.3%, but its relatively large model size (approximately 35.8 million parameters) restricts its suitability for lightweight deployment. In contrast, Wang et al. [28] designed the lightweight YOLO-PEL detector, which significantly reduces model size (only 2.23 million parameters), but its detection accuracy (mAP@0.5 of 32.5%) remains insufficient for practical applications. Among Transformer-based small object detection methods, Wei et al. [29] refined RT-DETR to develop DV-DETR, attaining an mAP@0.5 of 50.2% with 19.5 million parameters, demonstrating improved performance; however, limitations remain in feature alignment and contextual information fusion. Similarly, Wei and Wang [30] introduced RFAG-YOLO, which integrates receptive field attention and achieves 38.9% mAP@0.5 with only 5.9 million parameters, indicating that lightweight models still face challenges in accuracy. Wu et al. [31] combined HRFPN and EfficientVMamba techniques to achieve an mAP@0.5 of 38.9% with 33.5 million parameters under complex aerial scenarios, demonstrating strong detection capabilities, albeit with relatively heavy computational costs that restrict real-time applicability. Kong et al. [32] proposed Drone-DETR, an enhanced RT-DETR variant optimized for small object detection in remote sensing images, which achieves the highest mAP@0.5 of 53.9% and maintains an inference speed of 30 FPS. However, its computational complexity is as high as 128.3 GFLOPs, and it contains 28.7 million parameters, imposing stringent hardware demands that limit its practical deployment on UAV platforms. Baidya and Jeong [33] explored ConvMixer-based YOLOv5 heads to improve localization precision.

To address these limitations, the present study proposes UAV-DETR, which introduces three novel modules—CAS, SOEP, and CSAM—to enhance adaptive feature alignment and contextual perception. Under the premise of maintaining real-time inference speed (30 FPS), UAV-DETR achieves a competitive mAP@0.5 of 51.6%, with a significantly reduced parameter count of 16.8 million and computational complexity of 71.4 GFLOPs. The comprehensive performance demonstrates that UAV-DETR achieves an excellent balance among detection accuracy, model compactness, and computational efficiency, thus showing strong potential for practical application in resource-constrained UAV small object detection tasks.

### 2.4. RT-DETR Model

RT-DETR, developed by the Baidu research team, represents a major advancement in the real-time object detection field. In addressing the high computational cost issues of the previous DETR, which prevented its application in real-time detection, RT-DETR is designed with a computationally efficient hybrid encoder composed of the Attention-enhanced Intra-scale Feature Interaction (AIFI) module and the cross-scale feature fusion (CCFF) module. The design in question decouples within-scale feature interactions and cross-scale fusion, thereby enabling efficient processing of multi-scale features. Furthermore, RT-DETR employs an IoU-aware query selection mechanism that optimizes the initialization of decoder queries. These innovations enable RT-DETR to capitalize on its non-post-processing requirement, rendering it suitable for real-time object detection. The RT-DETR model is illustrated in Figure 1.

However, directly applying RT-DETR to object detection in drone imagery presents some challenges. Due to the multiple downsampling operations in the model’s backbone network, feature loss occurs during the downsampling process, leading to missed or incorrect detections of small objects. Furthermore, RT-DETR does not adequately address spatial misalignment and context mismatches during feature fusion, which negatively impacts the model’s overall performance.

## 3. Methodology

### 3.1. UAV-DETR Model

Given the shortcomings of RT-DETR in drone-based object detection, particularly its deficiencies in small object detection, feature information loss, and context misalignment. This paper proposes UAV-DETR, an enhanced multi-scale object detection framework specifically designed for drone-based imagery. Building upon the strengths of end-to-end detection, UAV-DETR proposes three key modules: CAS, SOEP, and CSAM, which include CFC and SFC. Collectively, these modules significantly elevate the model’s detection performance in complex aerial scenarios.

The CAS module improves the capability of the backbone network to detect multi-scale objects by refining the feature hierarchy and enhancing inter-channel interactions. The SOEP module strengthens the semantic representation of mid- and low-level features in the neck network, elevating small-target discrimination accuracy. The CFC and SFC modules calibrate and align features within the hybrid encoder, improving cross-scale feature integration. These improvements make UAV-DETR more accurate and real-time in its drone-based remote sensing object detection tasks. The UAV-DETR model is illustrated in Figure 2.

### 3.2. CasNet

In datasets like VisDrone-2019, which consist of low-altitude UAV imagery, targets are often extremely small—ranging from just a few to a few dozen pixels—and the background is filled with dense urban textures, glare, and motion blur. Traditional CNN backbones, relying on fixed receptive fields, struggle to balance global context modeling with fine-grained local detail extraction essential for identifying such minuscule objects. To address this, we replace the BasicBlock structures in the backbone with Convolutional Additive Self-attention (CAS) blocks [34]. Each CAS block first uses a stack of three 3 × 3 depthwise separable convolutions to integrate local neighborhood features while implicitly encoding positional information critical for precisely locating sub-20 px targets. This is followed by the Convolutional Additive Token Mixer (CATM), which generates attention weights along both spatial and channel dimensions using lightweight convolution-Sigmoid operations. The spatial component suppresses background clutter and highlights regions containing small objects, while the channel component amplifies discriminative features against complex backgrounds. These weights are fused additively to recalibrate subsequent features dynamically. The entire process avoids softmax normalization and large matrix multiplications, keeping inference complexity on par with standard convolutions while enabling long-range spatial and cross-channel interactions in a single forward pass. This long-range modeling leverages contextual cues to reinforce ambiguous small targets, and the additive fusion preserves subtle features often lost in Softmax-based mechanisms. As a result, the network produces more accurate responses to small and occluded targets while conforming to the computational constraints of mobile and edge computing platforms.

To meet the computational and deployment requirements, we can adopt simpler operations instead of relying on matrix multiplication—which introduces a significant inference overhead—or Softmax, which may pose challenges for deployment. Following this rationale, for a feature tensor of size *H* × *W* × *C*, information interaction can be decoupled into the spatial domain (*N* = *H* × *W*) and the channel domain (C). Notably, prior works on scale decomposition have also explored using multiple heads along the channel dimension to enable parallel computation, as well as dividing spatial patches to accelerate processing. However, these methods essentially operate on the fundamental dimensions of the feature tensor.

The concept of spatial interaction is demonstrated in Figure 3, where the input *x* ∈ *R^H^*^*×W*×*C*^. Initially, local token information is aggregated using a 3 × 3 convolutional layer. Subsequent to this, the reduction of the channel dimension to 1 is achieved by employing a 1 × 1 convolutional layer; subsequently, a Sigmoid activation function is applied to generate the feature-adaptive spatial attention map *x_s_* ∈ *R^H^*^×*W*×1^. The following formal description has been provided:*x_s_* = *Sigmoid*(*D*_1×1_(*D*_3×3,*ReLU*,*BN*_ (*x*))) ⊙ *x*(1)

For channel-wise interaction, we draw inspiration from SENet [35]; however, rather than employing channel reduction, we utilize a 1 × 1 convolution to directly capture and integrate inter-channel information:*x*_*c*_ = *Sigmoid*(*D*_1×1_(*P*_1×1_(*x*))) ⊙ *x*(2)
where *P* denotes the adaptive pooling and *D* is the group convolution layer, the group number is set to the channel number by default. Stacking these two operations yields the feature map after spatial and channel domain interactions, denoted as *Φ*(*x*).

This work introduces a novel similarity metric, defined as the summation of context scores between corresponding elements of *Q*, *K* ∈ *R^N^*^×*C*^:*Sim* (*Q*, *K*) = *Φ*(*Q*) + *Φ*(*K*)(3)
where Query, Key, and Value are generated via independent linear transformations, for example, *Q* = *W_q_x*, *K* = *W_k_x*, and *V* = *W_v_x*, it can be demonstrated that the context mapping function, denoted by *Φ*(*·*), contains the essential information interactions. The advantage of this generalization is that it is not restricted to the domain of manual context design; rather, it extends to the realm of convolutional operations, thereby facilitating implementation. It is anticipated that the network will learn and identify more valuable tokens through the execution of multiple interactions on both *Q* and *K*. The integration of these components is achieved through additive operations, a linear approach that circumvents the high complexity of matrix multiplication while retaining the efficacy in preserving useful information. Therefore, the following formulation can be used to express the output of CATM:*O* = Γ (*Φ*(*Q*) + *Φ*(*K*)) ⊙ *V*(4)
where Γ(·) ∈
*R^N×C^* denotes the linear transformation for integrating the contextual information.

### 3.3. SOEP Module

Conventional detection layers (P3–P5) exhibit limited efficacy in identifying diminutive UAV targets. A strategic remediation involves incorporating a higher-resolution P2 detection tier, but this increases computation and post-processing time. It is imperative that novel multi-scale feature fusion structures are developed with a view to detecting targets on UAVs. This paper proposes a computationally efficient small-target feature-enhanced fusion network (Figure 4). The network integrates the P2 feature layer (small target information) with the P3 layer after SPDConv processing. The CSP-OmniKernel module fuses features based on the CSP and OmniKernel [36] method.

#### 3.3.1. SPDConv

SPDConv [37] is a CNN module specifically designed to improve low-resolution image and small target detection. It reduces the information loss and enhances the capture of detailed features by replacing the traditional step-size convolution and pooling operations, enhancing model discrimination capability in processing small targets and low-resolution images. SPDConv is a CNN module specifically designed to improve low-resolution image and small target detection. It reduces the information loss and enhances the capture of detailed features by replacing the traditional step-size convolution and pooling operations, thereby enhancing the model’s performance in processing small targets and low-resolution images.

Specifically, SPDConv comprises a space-to-depth (SPD) transformation layer followed by a stride-1 convolutional layer. The SPD component partitions intermediate feature tensors of dimensions S × S × C through spatial decomposition, yielding sub-feature representations *f_x_*_,*y*_:(5)$$\left\{\begin{array}{l} f_{0,0}=X\left[0:S:s,0:S:s\right],f_{1,0}=X\left[1:S:s,0:S:s\right],\ldots \\ f_{s-1,0}=X\left[s-1:S:s,0:S:s\right];\\ f_{0,1}=X\left[0:S:s,1:S:\right],f_{1,1},\ldots ,\\ f_{s-1,1}=X\left[s-1:S:s,1:S:s\right],\\ \ldots \\ f_{0,s-1}=X\left[0:S:s,scale-1:S:s\right],f_{1,s-1},\ldots ,\\ f_{s-1,s-1}=X\left[s-1:S:s,s-1:S:s\right] \end{array}\right.$$

Here, the *s* refers to the downsampling factor. The generation of each sub-feature map is achieved by means of downsampling the original feature map, with the process being executed for each map using a specific scaling factor. To mitigate feature degradation and channel explosion from aggressive downsampling, a scaling factor of 2 optimizes the information–computation trade-off.

The architecture subsequently concatenates all sub-feature maps along the channel dimension, thereby forming a new feature map *X*. This process reduces the spatial dimensions while concomitantly increasing the number of channels. Finally, the transformed feature map *X* is further refined by the application of a non-strided convolutional layer. This layer employs learnable parameters to reduce the channel dimensionality, with the aim of mitigating information redundancy caused by the increased number of channels, while preserving as much discriminative feature information as possible.

#### 3.3.2. CSP-OmniKernel Module

The proposed CSP-OmniKernel module is designed based on the CSP architecture. The CSP architecture partitions the input feature map into dual pathways: a lightweight convolutional subnetwork processes one partition, while the complementary partition undergoes direct residual transmission to subsequent layers. The dual-path features are channel-wise concatenated, forming the input to the subsequent processing stage.

As illustrated in Figure 4, a convolutional layer partitions the input feature tensor into four distinct channels prior to OmniKernel processing. The transformed feature representation subsequently undergoes channel-axis concatenation with three complementary feature channels. The next step is a convolution to adjust the channel dimensionality. Parameters and computational costs are reduced, while the efficiency of feature extraction is enhanced. Consequently, both model training and inference are accelerated.

The OmniKernel network is comprised of three distinct branches: a global branch, a large-kernel branch, and a local branch. This architecture enables systematic acquisition of multi-scale feature representations spanning global to local contexts, thereby significantly improving detection efficiency for diminutive objects. The structural configuration of the OmniKernel network is schematically presented in Figure 5.

In the large-kernel branch, the input *X_Large_* undergoes three convolution operations across three separate channels: a 31 × 31 depthwise separable convolution to capture a large receptive field and two depthwise separable convolutions with kernel sizes of 31 × 1 and 1 × 31, respectively, to extract elongated contextual information.

In the inference stage of the global branch, input images of substantially higher spatial resolution than those used in training are processed. Consequently, a 31 × 31 kernel is inadequate in covering the global receptive field. In order to address this issue, a dual-domain processing strategy is adopted in order to enhance the global modelling capability of the global branch. The global branch incorporates two modules, the Dual-Domain Channel Attention Module (DCAM) and the Frequency-Based Spatial Attention Module (FSAM). These modules contribute to improved global feature modeling through dual-domain enhancement.

Given the input feature *X_Global_*, the DCAM Module first applies Frequency Channel Attention (FCA) to *X_Global_*:(6)$$X_{FCA}=IF(F(X_{Global})⊗W_{1\times 1}(GAP(X_{Global}))$$
where F and IF respectively denote the Fast Fourier Transform (FFT) and its inverse transform. The output of the FCA module is denoted by *X*_FCA_, the 1 × 1 convolutional layer by *W*_1×1_, and global average pooling by GAP. Element-wise multiplication is indicated by ⊗.

After undergoing Fourier transform processing, the global features, which have been globally modulated in the frequency domain, are subsequently input into the spatial channel attention (SCA) module:(7)$$X_{DCAM}=X_{FCA}⊗W_{l\times l}(GAP(X_{FCA}))$$
where *X*_DCAM_ is the output of the DCAM module, which only enhances the dual-domain features at the coarse channel granularity. Subsequently, *X*_DCAM_ is fed into the frequency-based spatial attention module to further refine the spectrum along the spatial dimension:(8)$$X_{FSAM}=IF(F(W_{l\times l}(X_{DCAM}))⊗W_{l\times l}(X_{DCAM})$$
where *X*_FSAM_ represents the output produced by the FSAM module.

The local branch applies only a 1 × 1 depthwise separable convolution to the input *X*_Local_, which is used for modulating local features.

### 3.4. CSAM Module

The hybrid encoder in RT-DETR falls short in cross-scale feature fusion: it does not adequately address spatial misalignment and fails to reconcile representational discrepancies across feature levels, which diminishes fusion effectiveness. Moreover, it overlooks the mismatch between pixel-level cues and their surrounding context, ignoring their distinct contextual dependencies and further constraining overall performance.

To address these shortcomings, we augment the CCFF module in RT-DETR’s hybrid encoder with two dedicated calibration mechanisms: Context Feature Calibration (CFC) and Spatial Feature Calibration (SFC) [38]. CFC harmonizes the semantic representations across different feature levels, while SFC rectifies the spatial misalignment introduced during multi-scale fusion, jointly improving feature alignment and overall information integration.

#### 3.4.1. CFC Module

The contextual information feature calibration method can adaptively construct and optimize the semantic contextual representation for each pixel. Meanwhile, by applying sharpening operations to large-scale objects and conditionally preserving spatial details, this method effectively facilitates selective learning of local context, fundamentally alleviating the significant bias toward large objects in the captured contextual information.

For a given feature tensor *X* ∈ *R^C^*^×*H*×*W*^, a pyramid pooling operation is first applied to extract multi-scale contextual representations *Z* ∈ *R^C^*^×*M*^. Subsequently, the spatial affinity between pixels and contextual features is computed via a softmax function, generating a spatial attention map Θ ∈ *R^N^*^×*M*^. Here, *C*, *H*, and *W* respectively denote the channel depth, height, and width of the input feature, while *N* = *H* × *W* and *M* represent the total number of pixels and contextual elements.

Contextual feature calibration (CFC) implements the aggregation of contextual information for individual pixels under the guidance of a spatial attention map. Subsequently, the response characteristics of each semantic context vector undergo refinement to generate a fine-grained context representation, effecting a recalibration of the contextual information. The complete CFC process is formulated in Equation (9).(9)$$y_{i}=\alpha_{i}\cdot \sum_{j=1}^{M}f(x_{i},z_{i})\cdot z_{j}+x_{i}$$

In the formula, {$x_{i}$, $y_{i}$,$\;\alpha_{i}$, $z_{j}$} ∈ *R^C^*^×1^ represents input, output, recalibration factor, and context information, respectively. The value range of *i* is $\left[1,\ldots ,H\times W\right]$, *f*(·) is a function to calculate the similarity between features. The architecture of the CFC module is depicted in Figure 6.

Where CPP stands for Cascading Pyramid Pooling Module and CRB stands for Context Recalibration Module.

#### 3.4.2. SFC Module

To address the spatial detail degradation caused by repeated downsampling operations, existing methods [39] predominantly employ cross-level feature interaction mechanisms that enhance the semantic representation capacity of deep features by integrating fine-grained information from shallow high-resolution features. The feature fusion architecture adopted in the RT-DETR model also follows this technical paradigm. However, in-depth analysis reveals significant limitations in this coarse-grained cross-scale feature aggregation approach: First, the discrepancy in receptive fields among different hierarchical feature maps leads to mismatches in spatial coordinate systems, resulting in spatial heterogeneity of feature representations. Second, hierarchical features exhibit significant disparities in semantic abstraction levels, where direct feature superposition induces semantic confusion. Notably, conventional methods employing uniform channel-wise alignment [40] strategies neglect the differential semantic information distribution across channel dimensions. This homogenization process undermines the distinctive characteristics of feature channels [41,42], thereby constraining further performance improvements of models.

To mitigate these constraints, the SFC module partitions channel dimensions into multiple sub-features for individualized calibration while adaptively integrating gating mechanisms to fuse cross-scale features. As shown in Figure 7.

Given a low-resolution feature map $F_{l}\in R^{C_{l}\times H_{l}\times W_{l}}$ and a high-resolution feature map $F_{h}\in R^{C_{h}\times H_{h}\times W_{h}}$, the SFC module first aligns their channel dimensions via convolutional layers. The low-resolution feature $F_{l}$ is then upsampled through bilinear interpolation and concatenated with $F_{h}$. Subsequently, these are fed into convolutional blocks to predict two sets of offset feature maps, ${∆}_{l}\in R^{\left(2\times G\right)\times H\times W}$, ${∆}_{h}\in R^{\left(2\times G\right)\times H\times W}$ for aligning the two feature levels, as well as two gating masks, *β_l_* and *β_h_*, which control the flow of feature information across the two scales. The calibrated features from both scales are ultimately fused via element-wise summation, with the complete operational procedure formalized in Equation (10).*O* = *β*_*l*_ ⊗ *T* (*V* (*W*_*l*_
*F*_*l*_), Δ_*l*_) + *β*_*h*_ ⊗ *T* (*W*_*h*_
*F*_*h*_, Δ_*h*_)(10)

Here, *V*(·) denotes the bilinear upsampling function, and *T*(·, Δ) represents a deformable sampling operator that adaptively adjusts spatial sampling locations based on learned offsets Δ. When Δ = 0, the operation becomes standard bilinear interpolation, resulting in an identity transformation under the gating mask *β* = 1 + tanh(0) = 1. When Δ ≠ 0, the offsets introduce spatial modulation, enabling the module to realign features across resolutions and correct spatial misalignments in a content-adaptive manner. *W_l_* and *W_h_* represent convolutional layers followed by Batch Normalization (BN) and ReLU activation. In addition, the SFC module adopts the 1 + tanh activation function for the gating masks. Thus, for the initial state *T*(*F*,0), the transformation corresponds to an identity mapping due to *β* = 1+ tanh(0) = 1. Under this condition, the SFC operation can be simplified and is expressed as shown in Equation (11).(11)$$O=V\;(W_{l}\;F_{l})+W_{l}F_{h}$$

## 4. Results and Analysis

### 4.1. Experimental Dataset and Experimental Setup

#### 4.1.1. Experimental Dataset

This study utilizes the publicly available VisDrone2019 dataset [43] for model training and evaluation. Containing drone-captured aerial footage, the dataset spans heterogeneous traffic environments (urban/rural) under diverse meteorological and photometric conditions. It contains 6471 training images, 548 validation images, and 1610 test images, with a significant prevalence of small-scale objects that facilitates rigorous evaluation of drone-based detection models. The proposed model demonstrates effectiveness in detecting ten object categories: van, tricycle, car, motor, awning-tricycle, bicycle, pedestrian, truck, bus, and people.

#### 4.1.2. Experimental Setup

All architectural modifications and training experiments were performed under identical computational conditions, with detailed specifications provided in Table 1

The experimental network training parameters are enumerated in Table 2.

### 4.2. Evaluation Metrics

To comprehensively evaluate the efficacy of the proposed enhancements, three critical metrics were employed: parameter count, Floating Point Operations (FLOPs), and Mean Average Precision (*mAP*). The mathematical formulation is defined as*P* = *TP*/(*TP* + *FP*)(12)*R* = *TP*/(*TP* + *FN*)(13)(14)$$AP=\int_{0}^{1}P(R)dR$$(15)$$mAP=\sum_{i=1}^{C}AP_{i}/C$$
where *TP* refers to the positive samples that have been predicted to belong to the positive category, *FP* denotes the negative samples that have been predicted to belong to the positive category, *FN* refers to the positive samples that have been predicted to belong to the negative category, *P* represents precision, and *R* denotes recall. *AP* (Average Precision) represents the average precision for a single class. The mean of the *APs* of the entire category, *mAP*, is influenced by the IoU threshold and is subsequently utilized at mAP@0.5 and mAP@0.5:0.95.

### 4.3. Results and Comparsion

#### 4.3.1. Ablation Study

To quantify component-specific contributions, we conducted systematic ablation studies by sequentially integrating CasNet, SOEP, and CSAM modules into the baseline RT-DETR framework. Training and validation utilized the VisDrone2019 dataset, with experimental results demonstrating progressive performance gains at each integration stage. Quantitative ablation results are presented in Table 3.

The introduction of CasNet resulted in a substantial enhancement of mAP@0.5 (from 48.1% to 49.2%) and mAP@0.5:0.95 (from 29.5% to 30.3%), concurrently reducing both parameters and FLOPs. This outcome serves to substantiate the hypothesis that CasNet is characterized by a lightweight and efficient design. The incorporation of the SOEP module led to a substantial enhancement in recognition precision (mAP@0.5 to 50.7%, mAP@0.5:0.95 to 31.5%), primarily resulting in a significant boost in recall, though this was accompanied by an increase in computational load. The final integration of CSAM yielded the most optimal overall results (mAP@0.5 of 51.6%, mAP@0.5:0.95 of 32.1%). The total computational cost of UAV-DETR increases to 71.4 GFLOPs, which remains within an acceptable range and is suitable for real-time UAV applications.

These findings confirm that each module—CasNet, SOEP, and CSAM—contributes distinct and complementary improvements, jointly enhancing detection accuracy and robustness in UAV-based tiny object detection tasks.

#### 4.3.2. Comparative Experiment

This paper selects several mainstream backbone networks in recent years—FasterNet [44], EfficientViT [45], and the original RT-DETR backbone ResNet18—for comparative experiments. By evaluating their overall floating-point operations (FLOPs) and detection accuracy on the VisDrone2019 dataset, we aim to identify the most suitable backbone network for object detection in drone imagery.

During the experiments, all models employing different backbone networks were trained and evaluated under identical hyperparameter settings. The training results for each backbone network are summarized in Table 4.

After conducting comparative experiments with mainstream feature-extraction networks, CasNet was selected as the backbone because it delivers superior feature-extraction performance on UAV imagery while requiring fewer floating-point operations than ResNet-18.

To rigorously benchmark UAV-DETR against state-of-the-art methods in drone-based detection, comparative experiments were conducted with mainstream detectors (Faster R-CNN, YOLO series) on the VisDrone2019 dataset. All models underwent consistent hyperparameter configuration during training and evaluation. Performance was quantified using standard metrics, including mAP@0.5 and GFLOPs, with comparative results detailed in Table 5.

In terms of detection accuracy (mAP@0.5), UAV-DETR achieves a score of 51.6%, outperforming most comparison models (including Faster R-CNN, the YOLO series, LGI-DETR, and the RT-DETR baseline) while approaching the top performers Enhanced YOLOv7 (52.3%) and Drone-DETR (53.9%). Specifically, it improves by 11.9 percentage points over the traditional two-stage detector Faster R-CNN (39.7%), indicating a significantly stronger capability in feature extraction and object localization for small targets. Compared with the YOLO series, UAV-DETR surpasses YOLOv5l (38.8%), YOLOv6m (37.2%), YOLOv7 (48.0%), and YOLOv8m (43.2%) by 12.8%, 14.4%, 3.6%, and 8.4%, respectively, though it trails Enhanced YOLOv7 (52.3%) by 1.3 percentage points. Among other recent detectors, UAV-DETR exceeds LGI-DETR (46.0%) by 5.6 points, CFPT (50.0%) by 1.6 points, and DV-DETR (50.2%) by 1.4 points, while being 2.3 points lower than Drone-DETR (53.9%).

Regarding model complexity, UAV-DETR has the smallest number of parameters among all models, at only 16.8 million, which is 2.7 million fewer than DV-DETR (19.5 M), the next lightest model. Although Enhanced YOLOv7 and Drone-DETR achieve slightly higher accuracy, they come at the cost of significantly increased model complexity, with Enhanced YOLOv7 requiring 19 M more parameters and Drone-DETR requiring 11.9 M more parameters compared with UAV-DETR. In terms of computational cost, UAV-DETR requires 71.4 GFLOPs, significantly lower than CFPT (297.6) and Drone-DETR (128.3) and only slightly higher than RT-DETR (57.0), while still achieving 3.5 percentage points higher detection accuracy, highlighting a superior trade-off between efficiency and performance.

In summary, the proposed UAV-DETR demonstrates superior detection accuracy across drone-based object detection tasks, even with a relatively low model size. These results indicate that UAV-DETR is highly suitable for real-time, accurate object detection in UAV imagery.

#### 4.3.3. Generalization Experiments

To validate the generalization capability of the proposed UAV-DETR model, we conduct experiments on the DOTA dataset, which contains a wide variety of object categories and complex aerial scenes. Specifically, we compare UAV-DETR with several representative object detection models, including Faster R-CNN, YOLOv7, and RT-DETR. This evaluation helps assess the model’s performance in scenarios different from the VisDrone dataset, thereby demonstrating its robustness and adaptability across diverse UAV-based remote sensing environments.

Table 6 presents the experimental results of Faster R-CNN, YOLOv7, RT-DETR, and UAV-DETR on the DOTA dataset. As shown in the results, UAV-DETR achieves mAP@0.5 improvements of 7.0 and 2.1 percentage points over Faster R-CNN and RT-DETR, respectively, demonstrating the strong generalization capability of UAV-DETR.

#### 4.3.4. Feature Fusion Visualization Comparison Experiment

For enhanced visual interpretability of the model’s decision mechanisms, Grad-CAM++ was utilized to generate class-discriminative activation heatmaps. In the heatmap, color intensity is used to represent the density of feature point activations, with red corresponding to the highest concentration. As illustrated in Figure 8, a noticeably denser concentration of feature points around the target objects—rather than the background—is observed in the UAV-DETR model when compared with the baseline. This demonstrates that the UAV-DETR model responds more effectively to small objects in complex scenes, resulting in improved detection performance with fewer missed or false detections, as well as enhanced suppression of background interference.

#### 4.3.5. Comparison of Detection Results

Model detection performance can be qualitatively assessed through visualization. To conduct a targeted evaluation of small object detection capabilities, three challenging scenarios from the VisDrone2019 dataset were selected for comparative analysis between the UAV-DETR framework and its baseline RT-DETR counterpart.

As depicted in Figure 9, for the multi-scale object scenario depicted in Figure 9a, the UAV-DETR model demonstrates significantly better performance compared with the baseline model, particularly in densely populated regions with distant objects, where it exhibits notably higher detection accuracy. In the occlusion scenario shown in Figure 9b, the UAV-DETR model is able to accurately identify targets that are partially or completely obscured by trees, providing more complete and precise detection results. In the nighttime condition depicted in Figure 9c, the UAV-DETR model maintains stable performance and effectively identifies distant objects in low-light conditions.

From the detection results, it is evident that RT-DETR exhibits missed and false detections for small objects. In contrast, UAV-DETR is capable of detecting small targets that the original model fails to identify and also shows improved performance in detecting occluded objects. Moreover, it achieves this without compromising the detection of other targets, effectively balancing coarse and fine-grained information. The three proposed improvements significantly enhance the detection accuracy of RT-DETR in UAV aerial scenarios, thereby validating the effectiveness of the modifications.

## 5. Conclusions

This paper presents UAV-DETR, an enhanced Transformer-based object detection framework specifically designed for unmanned aerial vehicle (UAV) imagery. Built upon the foundation of RT-DETR, UAV-DETR integrates three well-established modules—Channel-Aware Sensing (CAS), Scale-Optimized Enhancement Pyramid (SOEP), and Context-Spatial Alignment Module (CSAM)—each improved to target specific challenges in UAV-based object detection. CAS is refined to enhance channel-wise feature sensitivity, thereby improving the detection of small and occluded objects. SOEP is optimized to mitigate feature degradation caused by repeated downsampling, enabling more robust multi-scale representation. CSAM is enhanced to address spatial and contextual misalignment across feature levels, ensuring better integration of spatial structure and semantic context. Through this targeted refinement and effective integration of the three modules, UAV-DETR significantly improves multi-scale perception and contextual understanding. Experimental evaluations on the VisDrone2019 dataset show that UAV-DETR obtains a mean average precision (mAP@0.5) of 51.6%, which represents a significant 3.5% improvement over the original RT-DETR model. Additionally, the model maintains low computational complexity and real-time inference speed, making it suitable for deployment in practical UAV-based applications. However, UAV-DETR also has some limitations. The detection performance is prone to degradation under certain conditions, such as fast object motion and adverse weather like haze. Future research will focus on further optimizing the model for diverse UAV scenarios and exploring adaptive real-time deployment strategies.

## Figures and Tables

**Figure 1 sensors-25-04582-f001:**
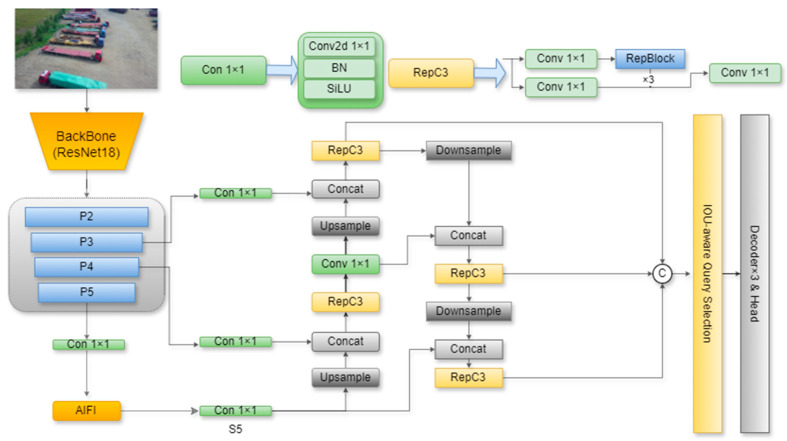
RT-DETR model structure.

**Figure 2 sensors-25-04582-f002:**
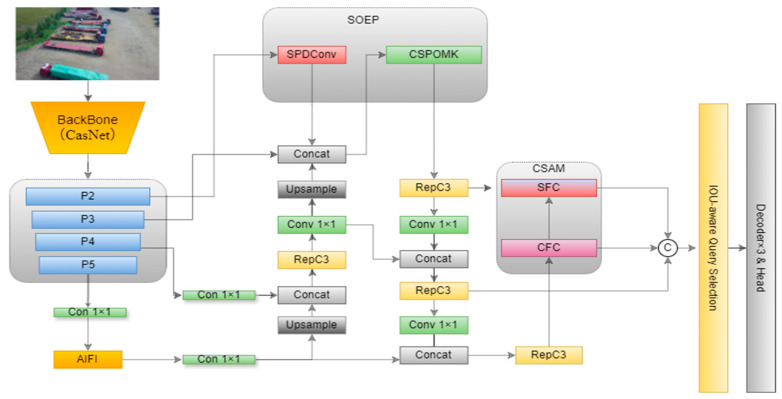
UAV-DETR model structure.

**Figure 3 sensors-25-04582-f003:**
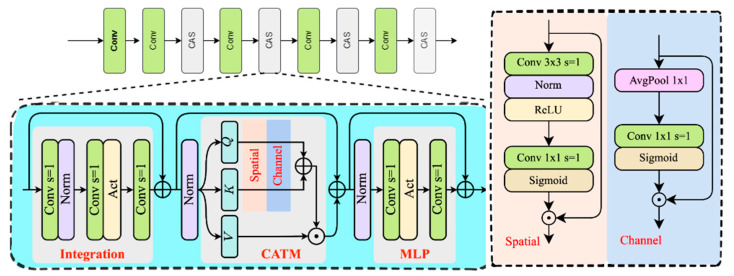
Architecture of CasNet.

**Figure 4 sensors-25-04582-f004:**
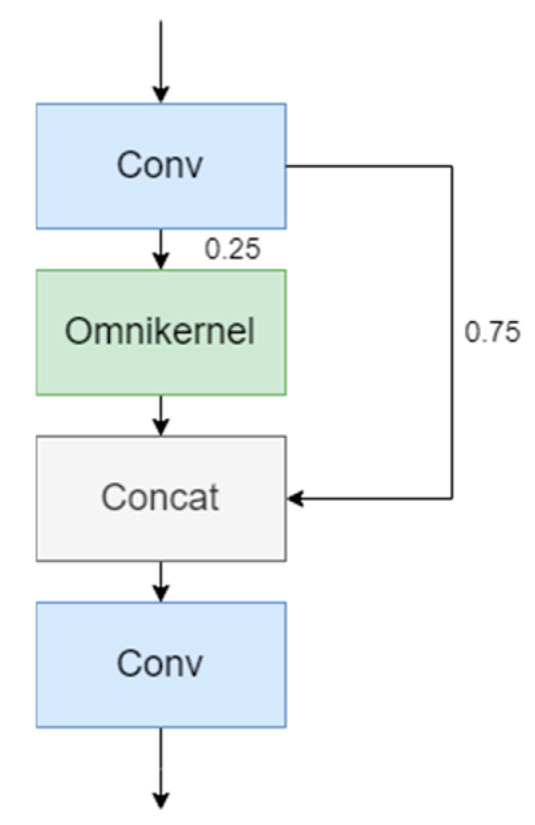
Architecture of CSP-OmniKernel.

**Figure 5 sensors-25-04582-f005:**
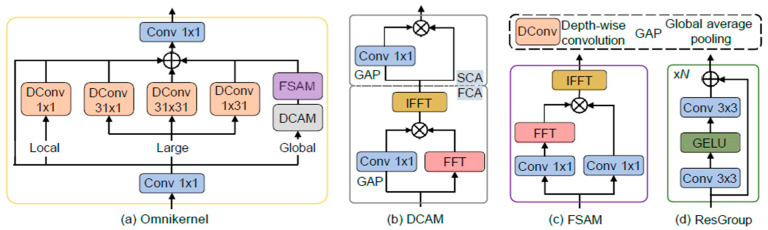
Architecture of OmniKernel.

**Figure 6 sensors-25-04582-f006:**
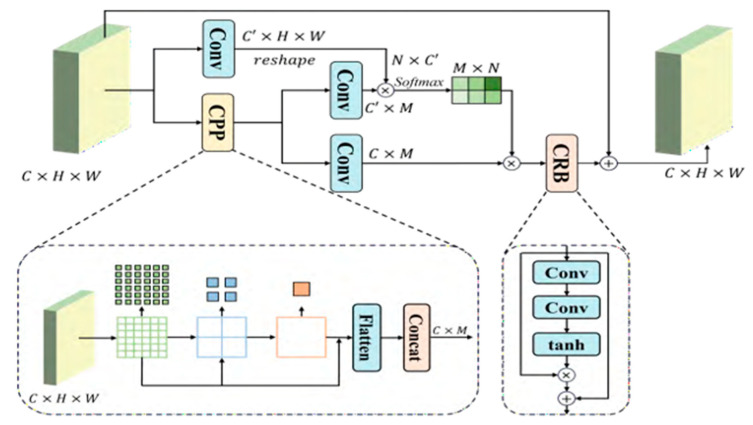
Architecture of CFC module.

**Figure 7 sensors-25-04582-f007:**
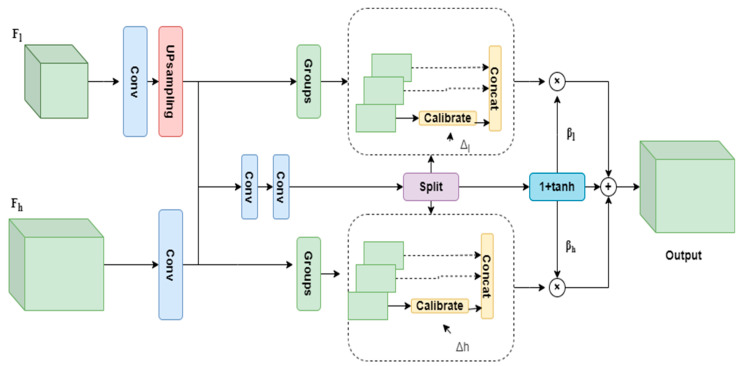
Architecture of SFC module.

**Figure 8 sensors-25-04582-f008:**
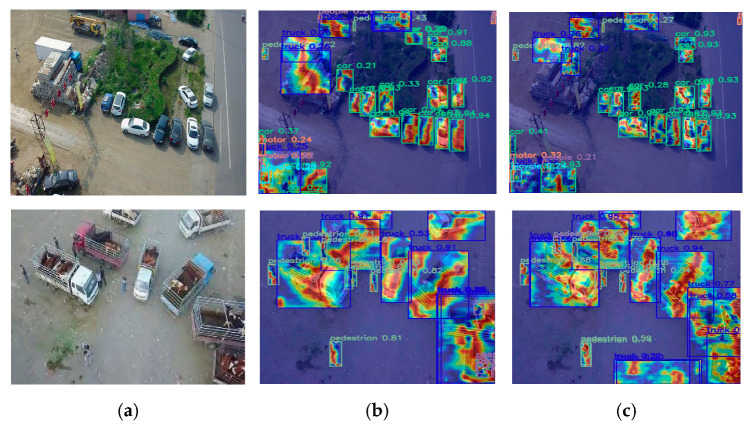
Comparison of heatmaps of original image, RT-DETR, and UAV-DETR model. (**a**) Original image. (**b**) RT-DETR. (**c**) UAV-DETR.

**Figure 9 sensors-25-04582-f009:**
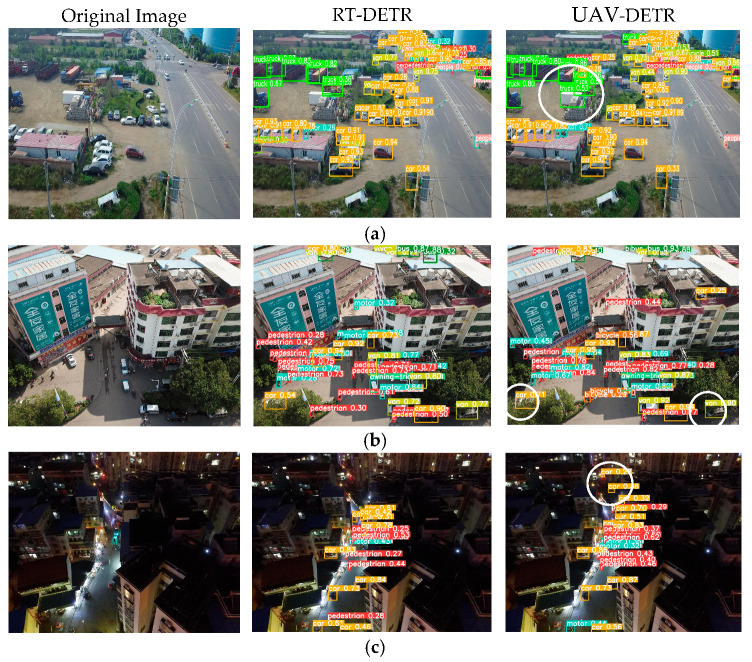
Comparative analysis of detection performance of original image, RT-DETR model, and UAV-DETR model: (**a**) multi-scale scenes; (**b**) occluded scenes; (**c**) nighttime scenes.

**Table 1 sensors-25-04582-t001:** Experimental environment.

Configurations	Parameters
OS	Windows 11
CPU	Intel(R) Core(TM) i9-10920X@3.50 GHz
GPU	NVIDIA GeForce RTX 3080 Ti 28 GB
CUDA Version	CUDA 12.1
Memory	32 GB
Deep Learning Framework	PyTorch 2.0.0

**Table 2 sensors-25-04582-t002:** Training parameters.

Name	Parameters
Epoch	500
Batch Size	4
Input Size	640 × 640
Initial Learning Rate	0.0001
Weight Decay Coefficient	0.0001
Optimizer	AdamW
Patience	30

**Table 3 sensors-25-04582-t003:** Ablation study results (%).

Number	Models	P	R	mAP@0.5	mAP@0.5:0.95	GFLOPs	FPS	Params/10^6^
Exp1	RT-DETR (Baseline)	61.8	47.0	48.1	29.5	57.0	45.8	19.8
Exp2	+CasNet	62.9	47.5	49.2	30.3	49.9	31.8	14.9
Exp3	+CasNet + SOEP Module	62.8	49.0	50.7	31.5	67.6	32.5	15.8
Exp4	+CasNet + SOEP Module + CSAM-Module	63.2	50.5	51.6	32.1	71.4	33.0	16.8

**Table 4 sensors-25-04582-t004:** Comparative Backbone Network Performance.

Backbone	mAP@0.5	GFLOPs/G
ResNet18	48.1	57.0
EfficientViT	47.9	52.6
FasterNet	47.6	54.9
CasNet	49.2	49.9

**Table 5 sensors-25-04582-t005:** Comparison of experimental results.

Model	mAP@0.5	Params/M	GFLOPs
Faster RCNN	39.7	41.4	212.9
YOLOv5l	38.8	46.1	107.8
YOLOv6m	37.2	34.8	85.6
YOLOv7	48.0	36.5	103.3
YOLOv8m	43.2	25.8	78.7
Enhanced YOLOv7 [26]	52.3	35.8	-
CFPT [24]	50.0	51.3	297.6
LGI-DETR [25]	46.0	21.1	65.0
DV-DETR [28]	50.2	19.5	84.6
Drone-DETR [31]	53.9	28.7	128.3
RT-DETR (Baseline)	48.1	19.8	57.0
UAV-DETR	51.6	16.8	71.4

**Table 6 sensors-25-04582-t006:** Generalization experiment results.

Model	mAP@0.5	Params/(M)	GFLOPs/G
Faster R-CNN	67.3	136.9	804.8
YOLOv7	74.5	36.6	103.4
RT-DETR	72.2	20.1	57.0
UAV-DETR	74.3	16.8	71.4

## Data Availability

The raw data supporting the conclusions of this article will be made available by the authors on request.

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
