# Peer review of "UAV-DETR: An Enhanced RT-DETR Architecture for Efficient Small Object Detection in UAV Imagery"

_sensors, 2025, doi:10.3390/s25154582_

Round 1
Reviewer 1 Report
Comments and Suggestions for Authors
Title: UAV-DETR: An Enhanced RT-DETR Architecture for Efficient Small Object Detection in UAV Imagery
This paper proposes UAV- DETR, an enhanced Transformer-based object detection model designed for aerial scenarios. Architecturally extending the RT-DETR framework, UAV-DETR incorporates three novel modules: Channel-Aware Sensing Module (CAS), Scale-Optimized Enhancement Pyramid Module (SOEP), and the newly designed Context-Spatial Alignment Module (CSAM). The work is good, and the authors have presented their work in detail; however, this manuscript currently contains major and minor corrections (shown below), which should be carefully considered.
Remarks to the Authors: Please see the full comments.
1- Based on the Abstract, please provide further clarification on the main point of the proposed work that addresses the feature degradation, and spatial-contextual misalignment in UAV-acquired imagery. On other word, what are the specific problems statement solved in the current work?
2- In the Abstract, there is no mention of any comparison with other recent related work in terms of the entire proposed work. This is an important point in every research paper to show the noticeable improvement compared to other work in the same field and to examine the performance. How was the performance examined? What is the improvement compared to other works in the same field?
3- The introduction section provides a foundation on the topic of this research based on various works; however, some other recent works need to be added to this section with much in-depth discussion by presenting the gaps that lead to propose this work. More discussion with supported references is required to show the robustness of DETR over CNN-based methods.
4- How does the Channel-Aware Sensing (CAS) module enhance small-target detection capability? Please explain.
5- It is recommended to enhance the section of the related work by providing previous recent works that are relevant to the current work and specifically related to the general idea of the proposed work and clarify their contributions, methodologies and limitations.
6- The sub section “3.1. RT-DETR Model” should be moved to the section of the “related work” and removed from the section of “Methodology” because “Methodology” section should contain the details of the proposed work.
7- A block diagram is required to illustrate the relationships between the various components or subsystems of the proposed work to simplify the reader understanding and break down components into simpler, interconnected blocks.
8- In general, the paper needs more organization and contains some grammatical and linguistic errors, and these errors lead to a decrease in the reader's understanding, so the entire manuscript should be reviewed carefully. In addition, the following notes should be addressed.
- The resolution of the figures should be enhanced, and Table 1 can be moved to section 4. -The Abbreviation should be defined in their first appearance.
-For Figure 8, please mention in the captions for subfigures “a and b”.
-Same note above for Figure 9.
-The font size should be unified. Please check line 529.
9- Any information, graph, equation, or data set taken from a previous source must be documented with a reliable source, unless it belongs to the authors. Please check this issue for the full manuscript. For example, the “Evaluation Metrics”.
14- comparative experiments with mainstream detectors (Faster R-CNN, YOLO series) have been made?
However, other comparative is required with recent related works to show the robustness of the full proposed work in terms of enhancing the feature degradation, and spatial-contextual misalignment.
15- For more readability, it is better to write the results obtained in Figure 9 as a table.
16- What is the rate of improvement achieved in terms of reduced computational complexity and real-time inference speed?
17- It is recommended to change the name of section “5” to Conclusion.
Comments on the Quality of English LanguageThe English should be improved to more clearly express the research.
Reviewer 2 Report
Comments and Suggestions for Authors
The authors proposed a number of architectural improvements for efficient small object detection in UAV imagery. Experimental results demonstrated that the proposed model outperforms existing object detection approaches on open access datasets. The paper presents an interesting approach, but there are several aspects that require refinement to improve the scientific soundness and clarity of the presentation.
1. The authors should expand the Related Work section by including existing works that address small object detection on the VisDrone2019-DET dataset. For example, the following papers also explore a similar problem:
- [Zhao, D.; Shao, F.; Yang, L.; Luo, X.; Liu, Q.; Zhang, H.; Zhang, Z. Object Detection Based on an Improved YOLOv7 Model for Unmanned Aerial-Vehicle Patrol Tasks in Controlled Areas. Electronics 2023, 12, 4887. https://doi.org/10.3390/electronics12234887]
- [Wang, Z.; Zhang, K.; Wu, F.; Lv, H. YOLO-PEL: The Efficient and Lightweight Vehicle Detection Method Based on YOLO Algorithm. Sensors 2025, 25, 1959. https://doi.org/10.3390/s25071959]
- [Wei, X.; Yin, L.; Zhang, L.; Wu, F. DV-DETR: Improved UAV Aerial Small Target Detection Algorithm Based on RT-DETR. Sensors 2024, 24, 7376. https://doi.org/10.3390/s24227376]
- [Wei, C.; Wang, W. RFAG-YOLO: A Receptive Field Attention-Guided YOLO Network for Small-Object Detection in UAV Images. Sensors 2025, 25, 2193. https://doi.org/10.3390/s25072193]
- [Wu, S.; Lu, X.; Guo, C.; Guo, H. Accurate UAV Small Object Detection Based on HRFPN and EfficentVMamba. Sensors 2024, 24, 4966. https://doi.org/10.3390/s24154966]
- [Kong, Y.; Shang, X.; Jia, S. Drone-DETR: Efficient Small Object Detection for Remote Sensing Image Using Enhanced RT-DETR Model. Sensors 2024, 24, 5496. https://doi.org/10.3390/s24175496]
- [Baidya, R.; Jeong, H. YOLOv5 with ConvMixer Prediction Heads for Precise Object Detection in Drone Imagery. Sensors 2022, 22, 8424. https://doi.org/10.3390/s22218424]
2. The authors should add relevant references for the mentioned modules. This is a good practice to highlight the authors' contribution. For example:
- the CAS block (Section 3.3) was introduced in [Zhang, T., Li, L., Zhou, Y., Liu, W., Qian, C., Hwang, J. N., & Ji, X. (2024). Cas-vit: Convolutional additive self-attention vision transformers for efficient mobile applications. arXiv preprint arXiv:2408.03703.]
- the module SPDConv (Section 3.4.1.) was introduced in [Sunkara, R., & Luo, T. (2022, September). No more strided convolutions or pooling: A new CNN building block for low-resolution images and small objects. In Joint European conference on machine learning and knowledge discovery in databases (pp. 443-459). Cham: Springer Nature Switzerland.]
- the Omni-kernel (Section 3.4.2.) was introduced in [Cui, Y., Ren, W., & Knoll, A. (2024, March). Omni-kernel network for image restoration. In Proceedings of the AAAI conference on artificial intelligence (Vol. 38, No. 2, pp. 1426-1434).]
3. It seems that the authors made a mistake with the abbreviation SRC in line 163.
4. In Figure 1, when explaining the RepC3 block, the authors refer to two different operations: Conv and Conv2d. They should clarify this ambiguity.
5. In Section 4.2, the authors provide a detailed description of precision (P) and recall (R) metrics. However, they do not describe the AP (Average Precision) metric, which is essential for calculating the mAP. It will be correct either to define P, R and AP or not to define all these metrics, just giving a relevant reference to the definitions.
6. There is a mismatch between Formula (10) and the explanatory Figure 7. Specifically, the formula applies interpolation to both feature maps; the text states that interpolation is applied only to the low-resolution feature map; the figure shows interpolation applied to the high-resolution feature map. The authors should carefully review Section 3.5.2 to ensure consistency between the text, equations, and figures.
7. The authors should define the operator T(·) in Formula (10) and explain the nature of its initial and non-initial states.
8. The authors should explain the parameters βâ‚— and βâ‚• in Formula (10).
9. The authors should clearly highlight their contributions in the Introduction and Summary sections, since most of the implemented architectural solutions already exist in prior work.
Reviewer 3 Report
Comments and Suggestions for Authors
Dear Authors;
- The manuscript addresses a highly relevant problem in UAV-based object detection, focusing on small object detection challenges. The proposed UAV-DETR model integrates novel modules CAS, SOEP, and CSAM and shows improved performance on the VisDrone2019 dataset.
- Some sections are overly technical and could benefit from clearer explanations or diagrams, particularly for readers not deeply familiar with Transformer-based models.
- While the integration of CAS, SOEP, and CSAM modules into RT-DETR appears innovative, the novelty claim should be more clearly justified in the context of prior Transformer-based UAV detection works. A more explicit comparison with similar recent models would strengthen this.
- Abstract
- The abstract is dense with technical terms. Consider simplifying it slightly to improve accessibility for a broader readership.
- It would help to briefly mention key performance improvements (e.g., mAP gains and model size reduction) in more plain language.
- Evaluation Metrics and Visualizations:
- Consider including mAP@0.75 or other IoU thresholds for more nuanced assessment.
- Grad-CAM++ visualizations are a nice touch. Could you include detection results on occluded or night-time drone imagery more clearly labeled?
- Efficiency Concerns:
- Although the model reduces parameter count, FLOPs slightly increase compared to RT-DETR. Could you clarify if UAV-DETR meets real-time constraints (e.g., frames per second) on typical UAV hardware?
- Dataset Limitation:
- Results are only reported on VisDrone2019. Did you attempt testing on UAVDT or other UAV-specific datasets to confirm robustness?
- Writing and Language:
- Several sections (especially in methodology) are highly technical. Consider breaking down some equations or referencing supplementary materials for deeper mathematical details.
- Improve grammatical flow in several areas for better readability.
- How does the UAV-DETR model perform on non-UAV datasets or in transfer learning settings?
- Are there any specific failure cases (e.g., occlusion, high speed movement, rain/fog conditions) where UAV-DETR underperforms?
- You mention adaptive real-time strategies in the conclusion—could you elaborate briefly on what techniques are under consideration?
Round 2
Reviewer 1 Report
Comments and Suggestions for Authors
Most of the comments have been addressed and no more are needed.
Comments on the Quality of English LanguageThe English could be improved to more clearly express the research.
Reviewer 2 Report
Comments and Suggestions for Authors
The authors proposed a number of architectural improvements for efficient small object detection in UAV imagery. As such, comments 1-5 and 7-9 have been adequately addressed. However, some issues remain unresolved.
Concerning the comment 6. In Formula (10), the interpolation operator V(·) is applied to both low resolution and high resolution feature maps. However, the text (lines 437-438) and Figure 7 indicate that interpolation is used only for the low-resolution feature map. So, the authors should correct Formula (10) to match the description in the text and Figure 7.
Also, the revised text has also generated the following additional comments:
1. The authors should provide a more detailed summary of the related works from references [24]–[31] in Section 2.3. Some of these papers propose approaches achieving higher mAP scores on the VisDrone dataset. Therefore, the authors should explicitly highlight the advantages of their method in comparison.
2. The authors should include the results of the mentioned related works in the comparative analysis (e.g., in Table 5). This would strengthen the paper’s scientific rigor, as direct comparison with state-of-the-art methods is a standard practice in high-quality journals.
3. Reference [10] does not match the citation in line 67. The authors should verify and correct this inconsistency.
4. The authors should complete the missing references in lines 133 and 136 (empty brackets). Proper citations are essential to support the claims and maintain academic integrity.
Reviewer 3 Report
Comments and Suggestions for Authors
Dear Authors;
All my suggestions have been addressed.
